
# Oscillations in atmospheric water above Switzerland

Klemens Hocke[1,2], Francisco Navas-Guzmán[1], Lorena Moreira[1], Leonie Bernet[1,2], and Christian Mätzler[1,2]

[1]Institute of Applied Physics, University of Bern, Bern, Switzerland
[2]Oeschger Centre for Climate Change Research, University of Bern, Bern, Switzerland

*Correspondence to:* K. Hocke
(klemens.hocke@iap.unibe.ch)

**Abstract.** Cloud fraction (CF), integrated liquid water (ILW) and integrated water vapour (IWV) were continuously measured from 2004 to 2016 by the TROpospheric WAter RAdiometer (TROWARA) at Bern in Switzerland. There are indications for inter-annual variations of CF and ILW while the IWV series of annual means mainly shows a positive linear trend. A spectral analysis gives the result that IWV is dominated by an annual oscillation leading to an IWV maximum of 24 kg/m$^2$ in July to

August and a minimum of 8 kg/m$^2$ in February. The seasonal behaviour of CF and ILW is composed by both, the annual and the semi-annual oscillation. However, the annual oscillation of CF has a maximum in December while the annual oscillation of ILW has a maximum in July. The semi-annual oscillations of CF and ILW are strong from 2010 to 2014. The normalized power spectra of ILW and CF show statistically significant spectral components with periods of 76, 85, 97 and 150 days. We find a similarity between the power spectra of ILW and CF with those of zonal wind at 830 hPa (1.5 km) above Bern. Particularly,

the occurrence of higher harmonics in the CF and ILW spectra is possibly forced by the behaviour of the lower tropospheric wind. The mean amplitude spectra of CF, ILW and IWV show increased short-term variability on time scales less than 40 days from spring to fall. We find a weekly cycle of CF and ILW from June to September with increased values on Saturday, Sunday and Monday.

## 1   Introduction

Observation and characterization of the oscillations of atmospheric water lead to a better understanding of the cloud processes, the cloud-induced changes in the Earth radiative fluxes, and the water cycle. In this study, we investigate the oscillations in 12-year long time series of cloud fraction (CF), integrated liquid water (ILW) and integrated water vapour (IWV) above Bern, Switzerland. The combined spectral analysis of atmospheric water parameters can give hints about cloud formation and transport processes. The seasonal cycle of the atmospheric water parameter CF at mid-latitudes was only described in a few

articles yet while the seasonal cycle in ILW seems to be undescribed yet. The climatology of IWV at Bern was presented by Morland et al. (2009) showing an annual oscillation with a summer maximum of about 22 kg/m$^2$ and a winter minimum of about 8 kg/m$^2$. This simple seasonal cycle in IWV is a consequence of the Clausius-Clapeyron equation and the seasonal cycle of air temperature at mid-latitudes.




Cossu et al. (2015) presented a 10-year cloud fraction climatology of liquid water clouds over Bern observed by the TROpo-spheric WAter RAdiometer (TROWARA). CF had a maximum of 60.9% in winter and a minimum of 42.0% in summer. They did not discuss the indication of a semi-annual oscillation in the seasonal cycle of CF. Hocke et al. (2016) divided the liquid water clouds into three classes: thin clouds, supercooled thick clouds and warm thick clouds using the TROWARA data set

5 at Bern. The warm thick clouds showed a CF maximum of 30% in the summer months and a minimum of 6% in winter. The CF of supercooled thick clouds was maximal in winter (29%) and minimal in summer (2%). Thin clouds had a fairly constant CF ranging from 30% in winter to 24% in summer. Massons et al. (1998) derived the seasonal cycle of cloud fraction using Meteosat images. CF was about 50% over the Iberian peninsula during winter and about 30% in summer

Compared to these few articles about the seasonal cycle of CF at mid-latitudes, there are more articles about the seasonal

10 change of CF over Antarctica, Arctic and the tropics. Meehl et al. (1998) described the mechanism of a semi-annual oscillation (SAO) in sea level pressure in the Southern Hemisphere which arises from different responses to the surface heat budget over the polar continent and the midlatitude ocean. van den Broeke (2000) investigated a possible relation between the SAO, the near surface wind and cloudiness. He found only at the Antarctic stations Halley and Faraday a firmly established half-yearly wave in the mean annual cycles of wind speed and cloudiness. Bromwich et al. (2012) gave a review about tropospheric

15 clouds in Antarctica. One focus was on the the seasonal and interannual variability of cloud amounts. Over the Southern Ocean equatorward of 60°S, only CloudSat-CALIPSO showed a minimum in cloudiness occurring in summer (5% lower than in winter). Verlinden et al. (2011), suggested that this summertime minimum is consistent with the seasonality of the extratropical cyclone activity.

Over the Arctic ocean, Beesley and Moritz (1999) compared observations and simulations of the seasonal cycle of the total

20 cloud amount. The observed seasonal cycle of CF is from 60% in winter to 85% in summer while the simulated seasonal cycle goes from 65% in winter to 75% in summer (if the simulation includes ice microphysics). The results of Beesley and Moritz (1999) suggest that the duration of the summertime cloudy season over the Arctic Ocean would be longer in a warmer climate and shorter in a cooler climate. The influence of wind speed on shallow marine cumulus convection was investigated by Nuijens and Stevens (2012). Their model simulations showed that an increase in the trade winds leads to a deepening of the

25 cloud layer.

For health and environmental reasons, the weekly cycle of aerosol concentration and precipitation is of high interest. Stjern (2011) detected weekly cycles in the $SO_2$ and $NO_2$ concentrations in the polluted region of the black triangle of Czech Republic, Germany and Poland. The weekly cycles of the $SO_2$ and $NO_2$ concentrations have decreased values at the weekend and increased values in the mid-week. The microphysical effect of the aerosol concentration on the formation and the size of cloud

30 droplets may induce weekly cycles in cloud parameters and precipitation. Another cause could be that the amount of aerosol concentration triggers surface diabatic heating and convective motions (Gong et al., 2007). Stjern (2011) found that weekly cycles of cloud amount and the frequency of light precipitation events above the Czech Republic are dominated by mid-week decreases and weekend maxima.

Our study extends the research on oscillations in atmospheric water by analysing the continuous measurements of the

35 TROpospheric WAter RAdiometer (TROWARA) at Bern, Switzerland. In section 2, we describe the ground-based microwave





radiometer TROWARA, its data set and the data analysis methods which we use in this study. Section 3 presents the seasonal cycles, the power spectra, and the bandpass-filtered annual and semi-annual oscillations in CF, ILW, and IWV. Inspired by the study of Nuijens and Stevens (2012), we look at the seasonal cycle and power spectrum of lower tropospheric wind which is provided by ECMWF operational analyses at the grid point close to Bern. Section 4 presents the climatologies of short-term

variability in CF, ILW, IWV and $u$ derived from daily means in the time interval from 2004 to 2016. We find a weekly cycle for CF and ILW in spite of the relatively clean air above Bern, Switzerland. Conclusions are given in Section 5.

## 2 Instrument, data and analysis

### 2.1 The microwave radiometer TROWARA

The study is based on the measurements of the TROpospheric WAter RAdiometer (TROWARA). TROWARA is a dual-channel

microwave radiometer built by (Peter and Kämpfer, 1992). It provides vertically-integrated water vapour (IWV) and vertically-integrated cloud liquid water (ILW), also known as liquid water path (LWP). TROWARA is located inside a temperature-controlled room on the roof of the EXWI building of the University of Bern (46.95°N, 7.44°E, 575 m a.s.l.). Since TROWARA is operated indoors, it is capable to measure IWV even during rainy periods.

The two microwave channels are at 21.4 GHz (bandwidth 100 MHz) and 31.5 GHz (bandwidth 200 MHz). The lower

frequency is more sensitive to microwaves from water vapour, and the higher frequency is more sensitive to microwaves from atmospheric liquid water.

The radiative transfer equation of a non-scattering atmosphere is

$$T_{B,i} = T_c e^{-\tau_i} + T_{mean,i}(1 - e^{-\tau_i}),\tag{1}$$

where $T_{B,i}$ the observed brightness temperature of the $i$-th frequency channel is (e.g., 21 GHz). $\tau_i$ is the opacity along the

line of sight of the radiometer, and $T_c$ is the contribution of the cosmic microwave background. $T_{mean,i}$ is the effective mean temperature of the troposphere (Ingold et al., 1998; Mätzler and Morland, 2009).

From equation 1 we can derive the opacities

$$\tau_i = -ln\left(\frac{T_{B,i} - T_{mean,i}}{T_c - T_{mean,i}}\right)\tag{2}$$

where the radiances $T_{B,i}$ are measured by TROWARA.

For a plane-parallel atmosphere, the opacity is closely related to IWV and ILW by a quasi-linear relationship

$$\tau_i = a_i'' + b_i'' IWV + c_i'' ILW,\tag{3}$$

where the coefficients $a''$ and $b''$ are not really constant since they can partly depend on air pressure. Mätzler and Morland (2009) show that these coefficients can be statistically derived by means of coincident measurements of radiosondes and fine-tuned at times of periods with a clear atmosphere. The coefficient $c$" is the mass absorption coefficient of cloud water. It

depends on temperature (and frequency), but not on pressure. It is derived from the physical expression of Rayleigh absorption





by clouds (Mätzler and Morland, 2009). Once the coefficients are determined, combined opacity measurements at 21 and 31 GHz permit the retrieval of IWV and ILW from equation 3. Thus a dual channel microwave radiometer can monitor IWV and ILW with a time resolution of 6-11 seconds and nearly all-weather capability during day and nighttime.

An infrared radiometer channel is operated at $\lambda = 9.5 - 11.5\mu$m which measures the physical temperature at the cloud base
when the cloud is optically thick (ILW $> 30$ g/m$^2$). TROWARA's antenna coil has a full width at half power of $4°$ and is pointing the sky at an zenith angle of $50°$ towards south-east. All the time, the view direction is constant, and the microwave and infrared channels of TROWARA observe the short-term temporal variations of the brightness temperature in the same volume of the atmosphere. This contributes to the high sensitivity of TROWARA for cloud detection. Further details of the sensors and retrieval technique are given in (Cossu et al., 2015) and (Mätzler and Morland, 2009).

TROWARA has been operated since 1994, and it has delivered an almost uninterrupted time series of ILW since 2004, with a time resolution of 11 seconds until end of 2009 and 6 seconds afterwards. The cloud detection in the line of sight of TROWARA is performed with the same time resolution, and the criterion is that ILW $> 3\sigma_{noise} = 2.3$g/m$^2$. Cossu et al. (2015) determined the instrumental noise $\sigma_{noise} = 0.77$g/m$^2$ of TROWARA from the noise of ILW during 245 days in which the sky was free of clouds. If a ILW value exceeds the $3\sigma_{noise}$ level, then we are confident by 99.7% that the ILW value was generated by a cloud
and not by instrumental noise. We emphasize that this is a remarkable sensitivity for a microwave radiometer. Contrary to the ILW series, the time series of IWV have been used since 1994 for trend analyses as has been shown by (Morland et al., 2009) and (Hocke et al., 2011)

Thin liquid water clouds were in the focus of the study by Hirsch et al. (2012). They derived the microphysical and optical properties of thin liquid water clouds and emphasized that these clouds should be considered in climate studies since these
clouds are frequent and they change the radiative forcing of the climate system. Measurements indicated that the downwelling infrared radiance of a thin liquid water cloud is increased by about 60% compared to clear sky. Hirsch et al. (2012) reported that thin liquid water cloud areas are often located at the edges of and in the inter-region between clouds (*twilight* zone of clouds).

Since TROWARA is not sensitive to ice clouds, CF of TROWARA is in general smaller compared to synoptic observations.
Cossu et al. (2015) found a CF difference of about 17% between TROWARA and synoptic observations in the same region over a period of 6 years. In addition, some of the very thin and tenuous clouds which are still visible by eye might be not seen by TROWARA. Hocke et al. (2016) derived CF of different classes of liquid water clouds using the TROWARA measurements and performed a trend analysis. In the present study, we only consider the class of all liquid water clouds with ILW $> 2.3$g/m$^2$. Finally, we like to mention that the CF, ILW and IWV measurements of TROWARA at Bern are representative for the Swiss
plateau. In the following, we investigate the monthly means of CF, ILW and IWV which we derived from the TROWARA data.

## 2.2 Data analysis

CF (cloud fraction) was determined in time domain. CF is the quotient of the time intervals when ILW $> 2.3$g/m$^2$ and the total observation time. The time intervals are as small as 6 seconds for ILW data after 2009 and 11 seconds for ILW data before 2009. Thus, we set the cloud flag with a high temporal resolution (6 or 11 second) which is required because of the high spatio-





temporal variability of clouds floating through the fixed line-of-sight of TROWARA. Monthly mean of ILW were obtained by averaging of the temporally high resolution data. An upper threshold of 400 g/m$^2$ is used that means in the presence of rain droplets we take the value 400 g/m$^2$ as an estimate of the ILW of the cloud droplets. During precipitation intervals TROWARA overestimates ILW of the cloud droplets because of the strong microwave emission from the rain droplets ($d > 0.2$ mm). This

is the reason, why we take an upper threshold of 400 g/m$^2$ for vertically integrated cloud liquid water path during rainy periods. Monthly means of IWV are well defined because of the continuous monitoring of IWV by TROWARA.

The power spectra are obtained by folding the time series of IWV, ILW or CF with a Hamming window and by applying zero padding at the beginning and end of the time series. After the Fourier transformation, the power spectra are normalized by the power of the strongest spectral component.

The time series of the annual oscillation (AO) and the semi-annual oscillation (SAO) are derived by means of bandpass filtering. The time series are filtered with a digital non-recursive, finite impulse response (FIR) bandpass filter performing zero-phase filtering by processing the time series in forward and reverse directions. The number of filter coefficients corresponds to a time window of three times the central period, and a Hamming window has been selected for the filter. Thus, the bandpass filter has a fast response time to temporal changes in the data series. The variable choice of the filter order permits the analysis

of wave trains with a resolution that matches their scale. The bandpass cutoff frequencies are at $f_c = f_p \pm 10\% f_p$, where $f_p$ is the central frequency. More details about the bandpass filtering are given by Studer et al. (2012).

The mean seasonal behaviour of the time series are obtained by sorting the data for the month and taking the mean and the standard error of the mean.

## 3   Long-term oscillations in atmospheric water with periods $> 60$ days

The time series of CF, ILW and IWV are shown in Figure 1. The green line corresponds to the monthly means while the red line is the 12 months-moving average. The blue lines denote the standard deviations of the parameter for an interval of 12 months. The annual cycle is only clear for the IWV series in the lower panel. The inter-annual variations (red line) of CF and ILW are quite similar. The red line of IWV shows a positive trend which is not subject of the present paper. The seasonal variations of CF and ILW are rather unclear. A spectral analysis of the green curves gives us more informations.

Figure 2 shows the normalized power spectra of the monthly mean series of CF, ILW, IWV and u. The horizontal red lines denote the two sigma-level where the confidence is 95%. The power spectrum of IWV is "most" simple. IWV has only one dominant annual oscillation. The power spectra of CF and ILW resemble each other to some extent and are prevailed by the annual and the semi-annual oscillation. Further there are statistically significant spectral components with periods of 76, 85, 97 and 150 days. For the interpretation of the CF and ILW spectra we add a power spectrum of the zonal wind at 830 hPa

(ca. 1.5 km altitude). The zonal wind series originates from ECMWF operational reanalysis at the grid point nearest to Bern (46.95°N, 7.44°E). It is surprising that the power spectrum of the zonal wind has strong annual harmonics reaching up to to the fourth harmonic. Since lower tropospheric wind is a major player for cloud formation and transport processes we suggest that the spectral components in the zonal wind spectrum are possibly the cause for the annual and semi-annual oscillations in





the power spectra of CF and ILW. In addition the periodicities of 97 and 85 days (close to the fourth harmonic) are strong in the spectra of u, CF and ILW.

Figure 3 shows the 12 month-bandpass filtered series of CF in the upper panel which corresponds to the annual oscillation. The amplitude of the annual oscillation was strongest around 2010 to 2011. The middle panel shows the semi-annual oscillation

which is obtained by means of a 6 month-bandpass filter. The semi-annual oscillation is strong from 2010 to 2014. The lower panel shows the combination of the AO and SAO (black line) which fits well to the unfiltered green line of the monthly means of CF. Figure 4 shows the bandpass filtered AO and SAO for the parameter ILW. Similar to CF, the SAO in ILW is strong from 2010 to 2014. The lower panel shows the combination of the AO and SAO (black line) which fits well to the unfiltered green line of the monthly means of ILW.

Figure 5 shows the climatologies of CF, ILW and IWV averaged over the time interval from 2004 to 2016. The left-hand-side panels show the mean AO (blue) and the mean SAO (red). It is surprising that the AO of CF is almost in anti-phase to the AO in ILW which peaks in July. We think that convective cumulus clouds are responsible for the high ILW values in July. The right-hand-side panels show the mean behaviour of the combined AO and SAO in black while the green lines show the mean behaviour derived from the monthly mean series of CF, ILW and IWV. In addition the standard error of the mean is given by

green error bars.

Figure 6 shows the climatology of eastward wind at 830 hPa (1.5 km) above Bern over the time from 2004 to 2016. It is obvious that the climatology of u is rather similar to the climatology of CF in Figure 5. It seems that the strong eastward wind in December and January transports stratus cloud layers to Switzerland. Related to the study of Nuijens and Stevens (2012) we may argue that an increase in the lower tropospheric wind u leads to a deepening of the cloud layer. In addition, one may

argue that an eastward advection of moist air from the Atlantic towards the Swiss plateau and the Alps occur which leads to a maximum of CF in winter.

## 4   Short-term oscillations in atmospheric water with periods $< 60$ days

For the investigation of the short-term variability, we change from the time series of monthly means to the time series of daily means. It can be assumed that the short-term oscillations with periods of a few days to weeks only persists over time intervals

of 3 wave cycles. Thus a Fourier transform over the time interval from 2004 to 2016 is not adequate to address the role of the short-term variability. Instead, we determine the mean amplitudes with a bandpass filter with a fast response time. As described in the data analysis section, the number of filter coefficients corresponds for each central frequency to a time interval of 3 wave cycles. Thus short-term variations existing over a short time interval contribute to the mean amplitude spectra which are shown in Figure 7. The amplitude spectra of CF, ILW, IWV and $u$ at Bern are derived by the wavelet-like bandpass filter method for

the time interval from 2004 to 2016. Again, $u$ originates from operational ECMWF reanalysis at 830 hPa (1.5 km) above Bern. The spectra of CF, ILW and $u$ are dominated by short-term variability on time scales less than 50 days. The amplitude maxima are at a period of 7 days for CF, 6 days for ILW, 365 days for IWV, and 17 days for $u$.





The bandpass filtered data sets are also appropriate for the derivation of the climatologies of CF, ILW, IWV and $u$. Figure 8 shows the mean amplitudes as function of the month and the period. The climatologies of CF, ILW and IWV show some similarities with increased amplitudes in the period range 5-10 days from spring to fall. The climatology of $u$ shows a 20 day-oscillation in winter which is possibly related to a Rossby wave. Now, we like to investigate if the 7 day-oscillation is

phase-locked to a weekly cycle which is found in aerosol concentration as induced by man-made air pollution (Gong et al., 2007). In the following, we only consider the data from 1 June to 30 September when the 7 day-oscillation is strong. Figure 9 shows a significant weekly cycle for CF and ILW, while the weekly cycle in IWV is marginal. The weekly cycles in CF and ILW have largest values on Sunday (day 1) and Monday (day 2) while the smallest values occur on Thursday (day 5). It remains an open question if the observed weekly cycles in CF and ILW are due to man-made air pollution. Barmet et al. (2009)

found a well-pronounced and statistical significant weekly cycle for particulate matter (PM) above Switzerland but they did not find a statistically significant weekly cycle for precipitation.

## 5  Conclusions

The TROpospheric WAter RAdiometer (TROWARA) continuously measured cloud fraction (CF), integrated liquid water (ILW) and integrated water vapour (IWV) at Bern in Switzerland from 2004 to 2016. We find indications for inter-annual

variations of CF and ILW while the IWV series of annual means mainly shows a positive linear trend. Fourier transformation and bandpass filtering give the result that IWV is dominated by an annual oscillation leading to an IWV maximum of 24 kg/m$^2$ in July to August. The seasonal behaviour of CF and ILW is composed by both, the annual and the semi-annual oscillation. However, the annual oscillation of CF has a maximum in December while the annual oscillation of ILW has a maximum in July. The semi-annual oscillations of CF and ILW are strong from 2010 to 2014. The normalized power spectra of ILW and

CF show statistically significant spectral components with periods of 76, 85, 97 and 150 days. We find a similarity between the power spectra of ILW and CF with those of zonal wind at 830 hPa (1.5 km) above Bern. The occurrence of higher harmonics in the CF and ILW spectra is possibly forced by the behaviour of the lower tropospheric wind. This observational result emphasizes the role of the lower tropospheric wind for generation and transport of clouds over the Swiss plateau. The climatology of CF shows a maximum in winter when the eastward wind is maximal. The mean amplitude spectra of CF, ILW

and $u$ are dominated by short-term variability on time scales less than 50 days. The short-term variability of CF, ILW and IWV has increased amplitudes from spring to fall. We find weekly cycles in CF and ILW for summer data (1 June to 30 September). The weekly cycles have largest values on Sunday and Monday. This result is consistent with Stjern (2011) who found that the weekly cycles of cloud amount and the frequency of light precipitation events are dominated by mid-week decreases and weekend maxima during summer. In difference to this observational result, Albrecht (1989) argued that increases in aerosol

concentrations may increase the amount of low-level cloudiness through a reduction in drizzle. The relevant mechanisms which lead to the observed weekly cycles in CF and ILW at Bern remain as an open question.



# 6 Code availability

Routines for data analysis and visualization are available upon request by Klemens Hocke.

# 7 Data availability

Hourly measurements of IWV and ILW from the radiometer TROWARA are available at the data centre STARTWAVE
5 (http://www.startwave.org) of University of Bern. 6-second-data of IWV, ILW and CF are available upon request by Klemens Hocke. We thank the European Centre for Medium-range Weather Forecast (ECMWF) for operational reanalysis data of zonal wind above Bern.

*Author contributions.* Klemens Hocke carried out the spectral analysis. Francisco Navas Guzmán and Christian Mätzler took care on the radiometer. All authors contributed to the interpretation of the data set.

10 *Acknowledgements.* The study was supported by Swiss National Science Foundation under grant number 200021-165516.



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





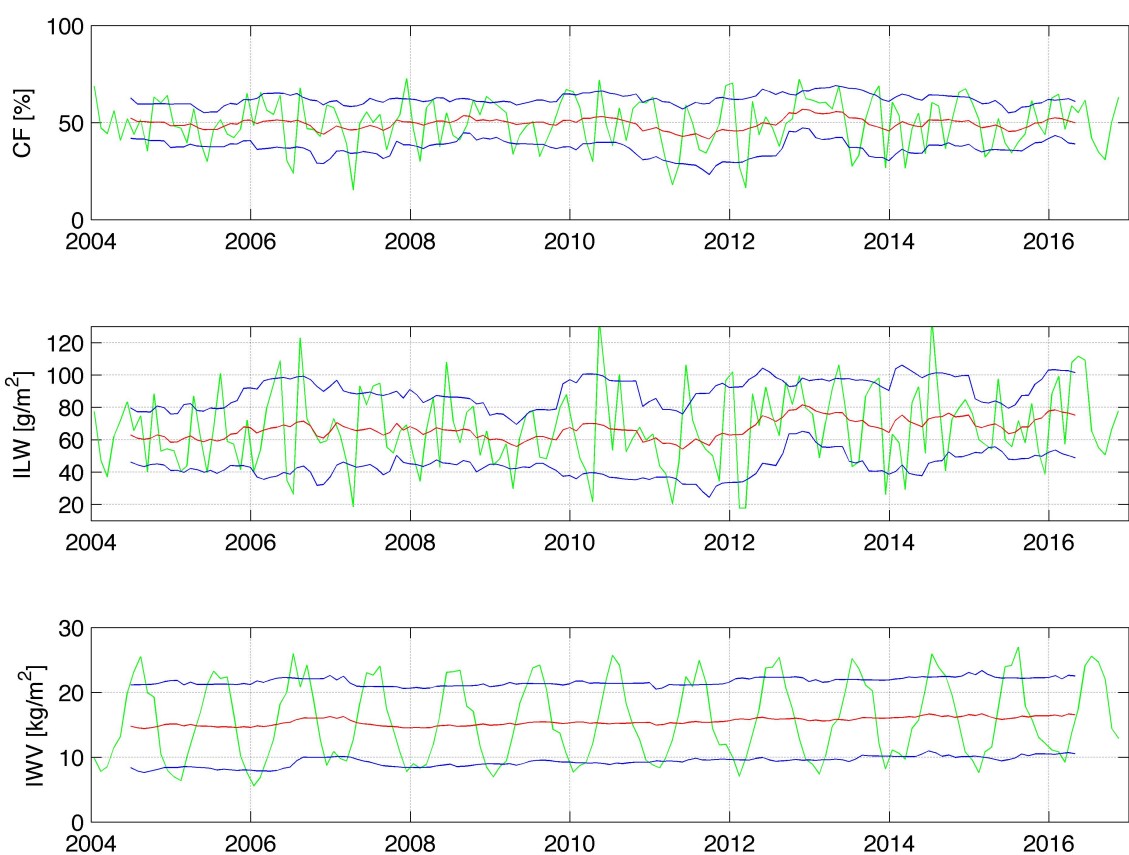

**Figure 1.** Time series of CF, ILW and IWV at Bern. The monthly means are given by the green lines while the red lines denote the annual means (12 months-sliding average with a step of 1 month). The blue lines shows the standard deviations of the annual means. Please note that a seasonal oscillation is only clear for IWV.





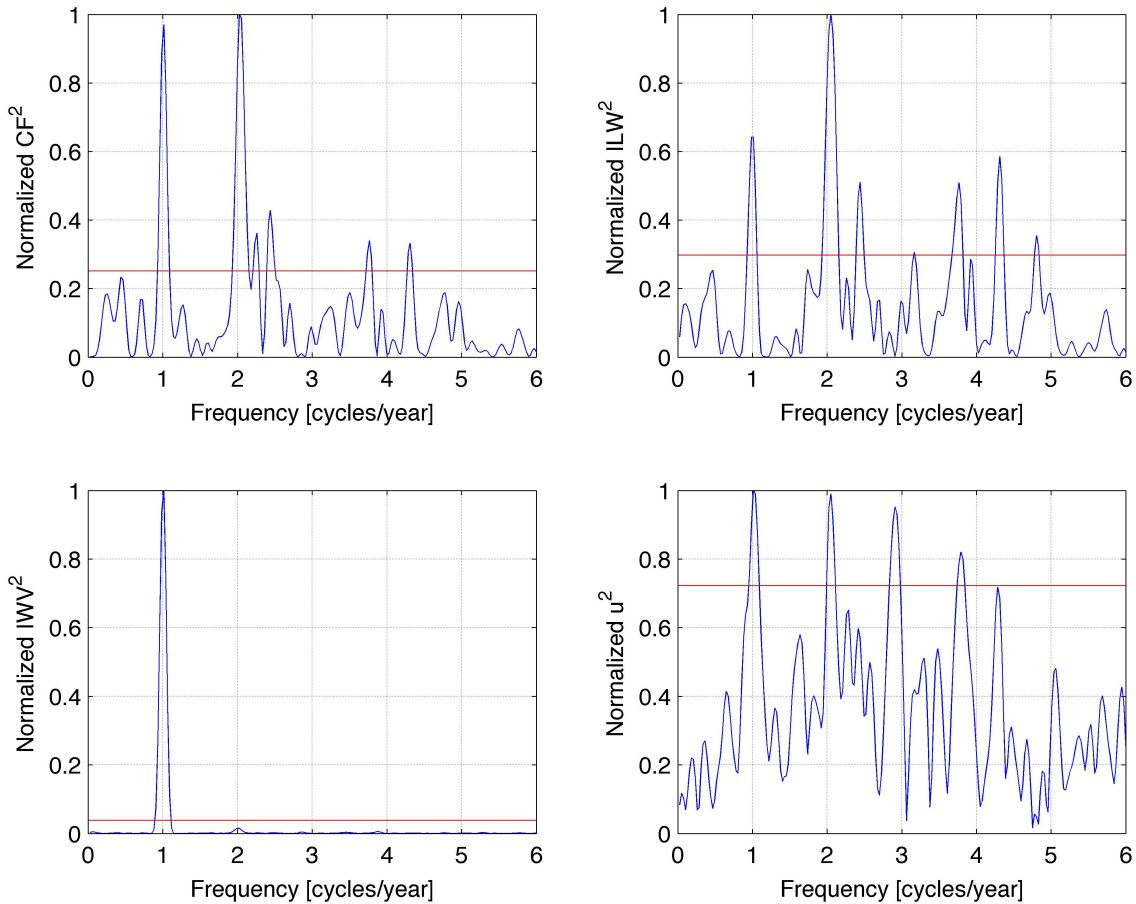

**Figure 2.** Normalized power spectra of CF, ILW and IWV at Bern for the time interval from January 2004 to November 2016. In addition we show the normalized power of the zonal wind $u$ from ECMWF operational reanalysis at 830 hPa (1.5 km altitude) above Bern and for the same time interval. The red line is the two sigma level (95% confidence). The semi-annual oscillation is approximately of the same size as the annual oscillation in case of CF, ILW and the zonal wind $u$.





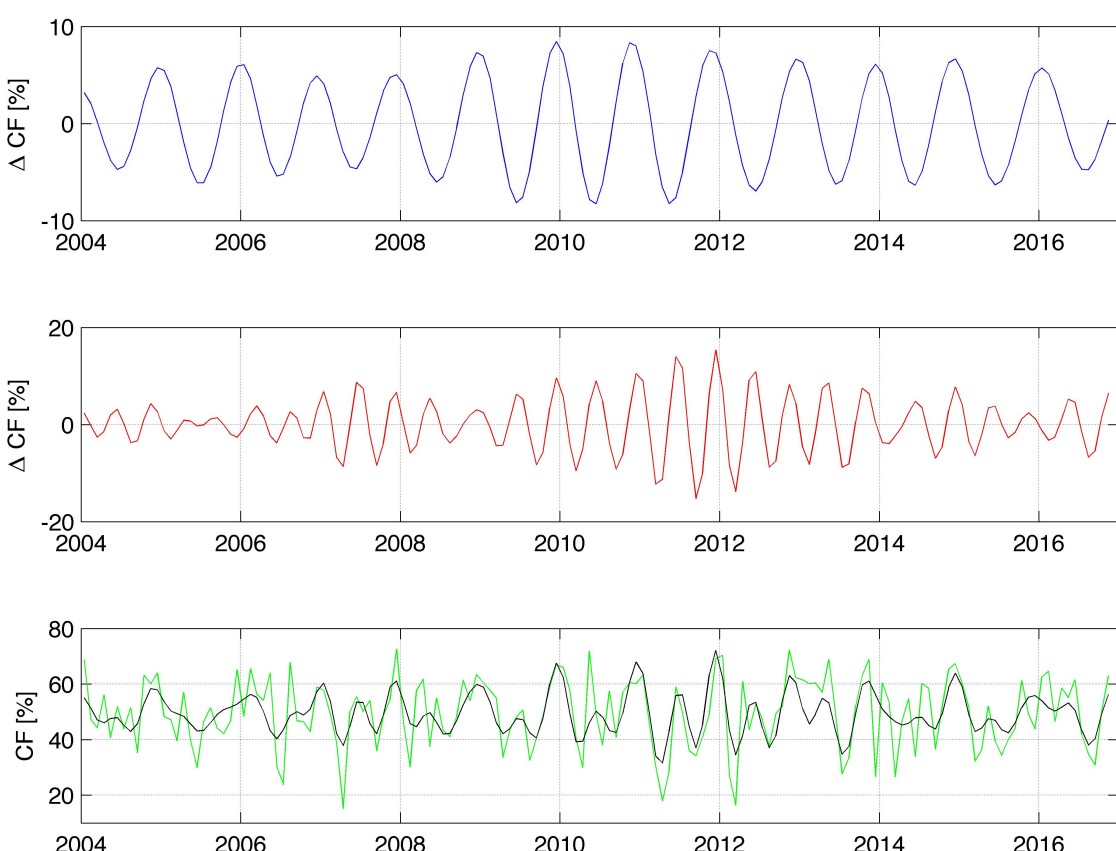

**Figure 3.** Annual oscillation (upper panel), semi-annual oscillation (middle panel) and time series of monthly means of CF (green line in the lower panel) derived from TROWARA measurements at Bern. The black line is the sum of the annual oscillation, the semi-annual oscillation and the total mean of CF.





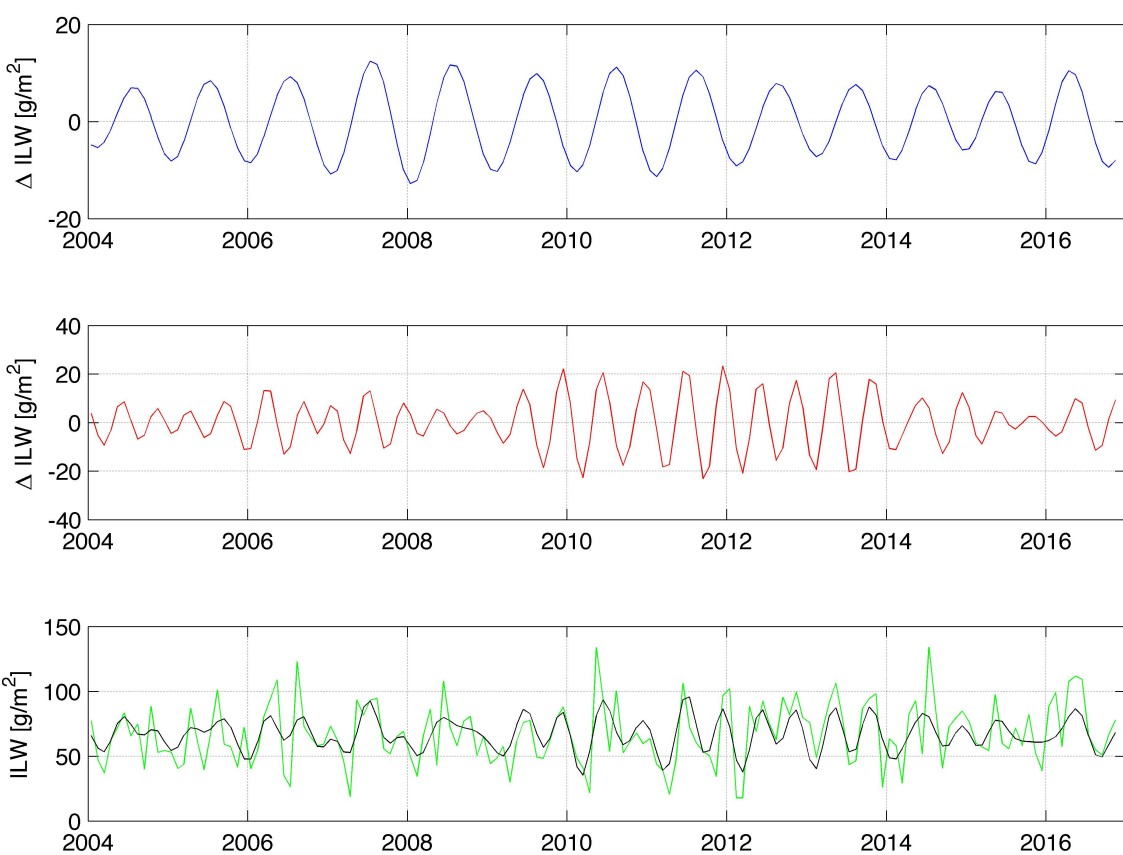

**Figure 4.** Annual oscillation (upper panel), semi-annual oscillation (middle panel) and time series of monthly means of ILW (green line in the lower panel) derived from TROWARA measurements at Bern. The black line is the sum of the annual oscillation, the semi-annual oscillation and the total mean of ILW.



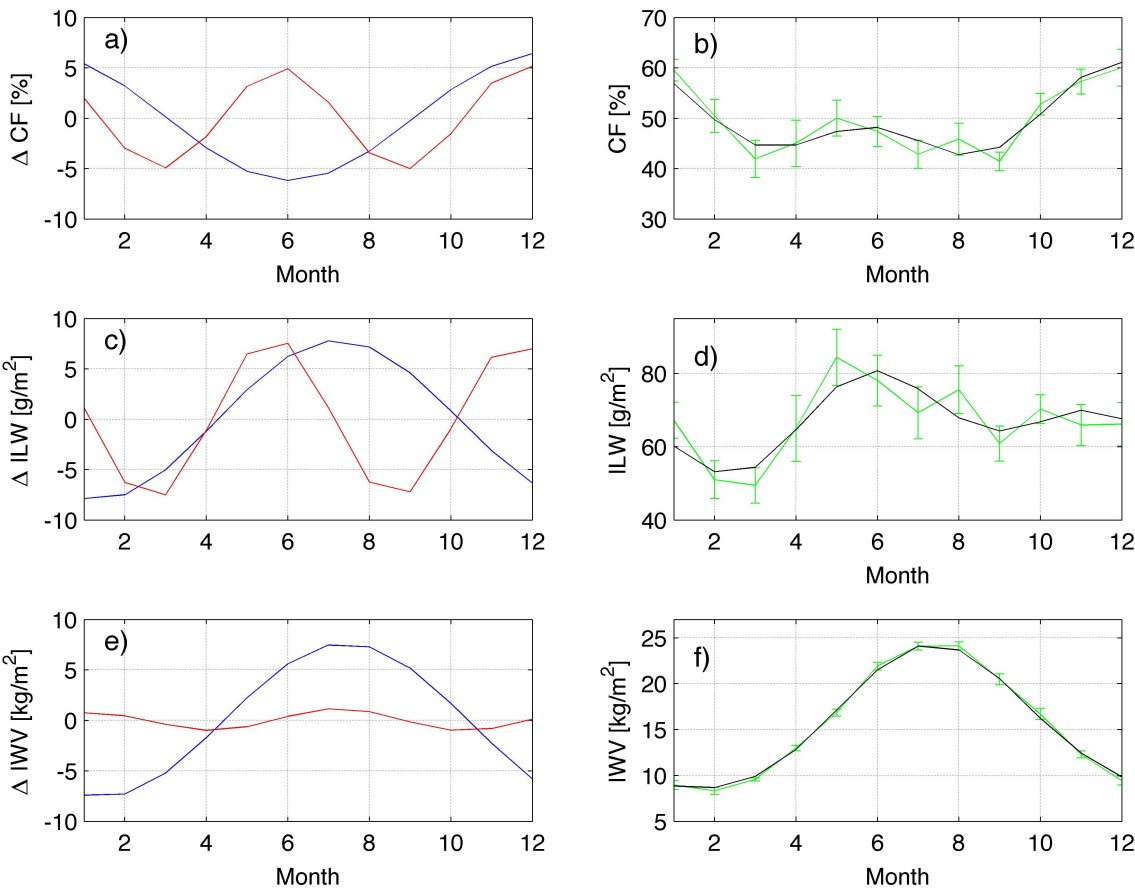

**Figure 5.** Mean seasonal behaviour of annual oscillation (blue), semi-annual oscillation (red), monthly means (green) and the sum of AO and SAO (black) derived from TROWARA measurements of the time interval 2004 to 2016. The top panels a) and b) are for CF, the middle panels c) and d) are for ILW and the bottom panels e) and f) are for IWV. The standard error of the mean is given by green error bars.




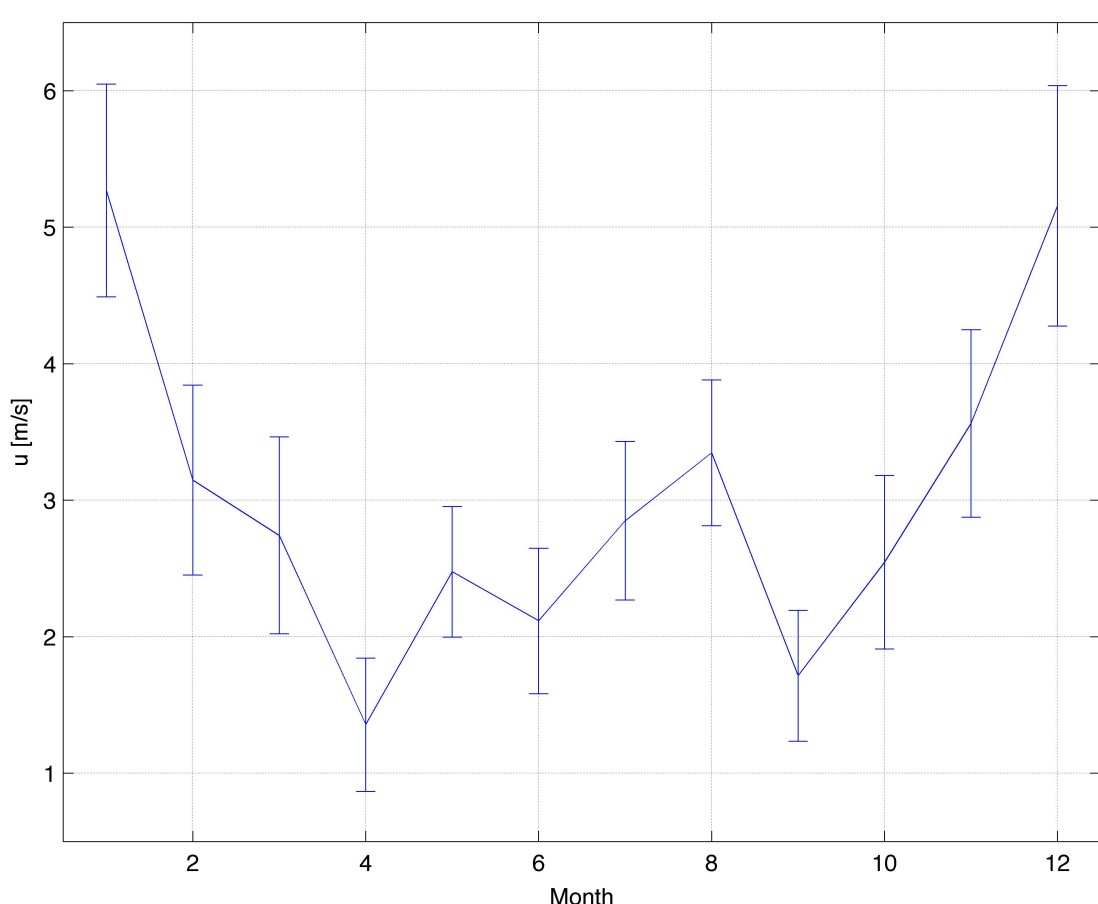

**Figure 6.** Mean seasonal behaviour of zonal wind $u$ from ECMWF operational reanalysis at 830 hPa (1.5 km altitude) above Bern. The seasonal change of $u$ is similar to that of cloud fraction in Fig. 5b). The standard error of the mean is given by error bars.





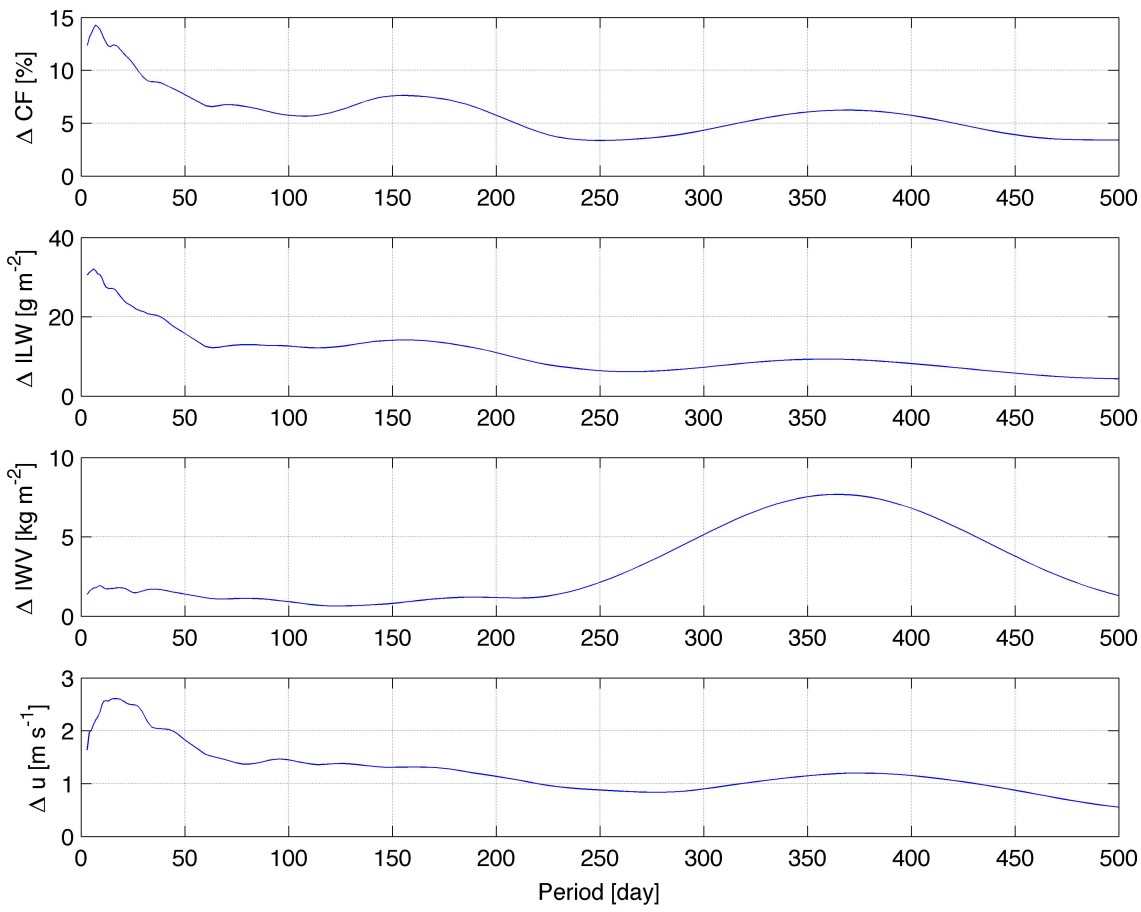

**Figure 7.** Mean amplitude spectra of CF, ILW, and IWV from TROWARA at Bern for the time interval from January 2004 to November 2016. In addition, we show the coincident amplitude spectrum of zonal wind $u$ from ECMWF operational reanalysis at 830 hPa (1.5 km altitude) above Bern. The amplitude is determined by a bandpass filter with a fast response time. The spectra of CF, ILW and u are dominated by short-term variability on time scales less than 50 days.

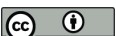



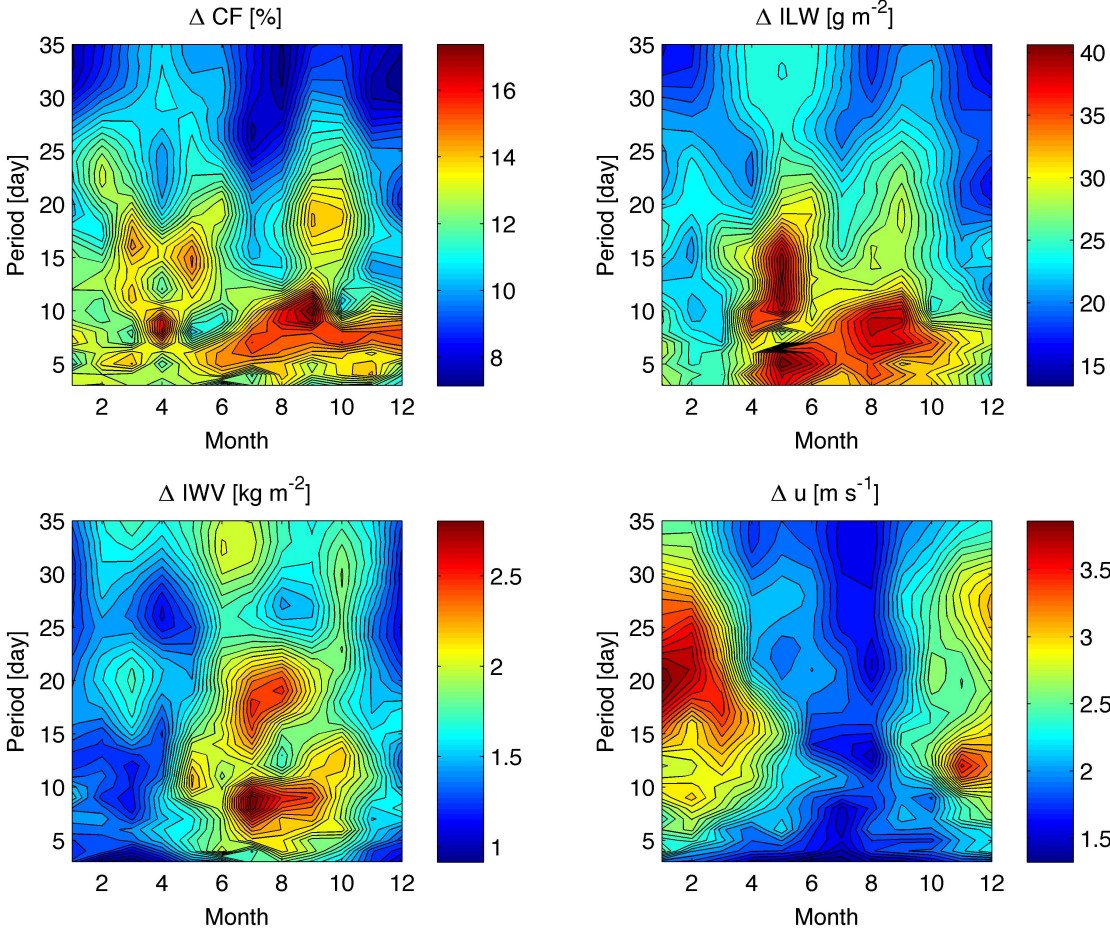

**Figure 8.** Climatologies of the short-term variability of CF, ILW, and IWV from TROWARA at Bern for the time interval from January 2004 to November 2016. In addition, we show the coincident climatology of zonal wind $u$ from ECMWF operational reanalysis at 830 hPa (1.5 km altitude) above Bern. The amplitudes of the short-term variations in CF, ILW and IWV are enhanced from spring to fall. There are indications of a summertime 7 day-oscillation in CF, ILW and IWV.




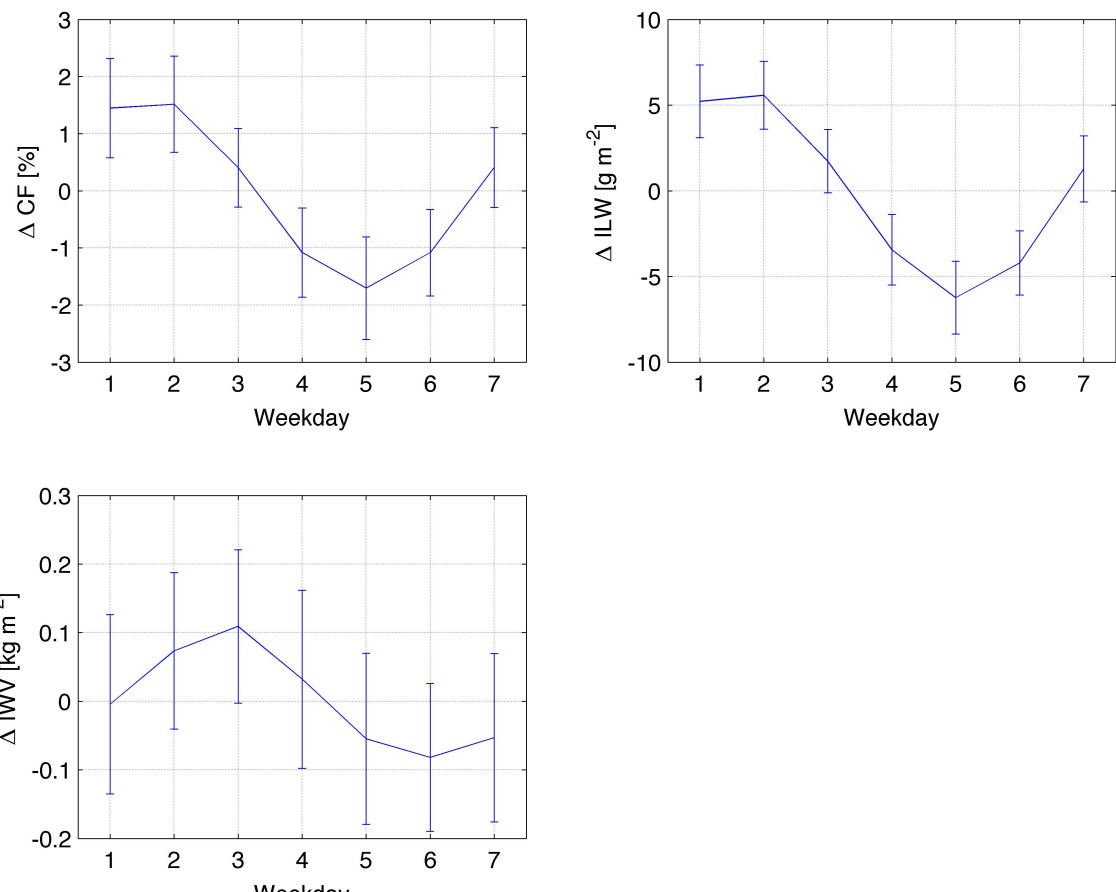

**Figure 9.** Weekly cycle of CF, ILW, and IWV at Bern for the June to September observations of TROWARA during the time interval from January 2004 to November 2016. While the weekly cycle in IWV is marginal, the weekly cycles in CF and ILW show largest values on Sunday (weekday 1) and Monday (weekday 2).