# Peer review of "Oscillations in atmospheric water above Switzerland"

_Atmospheric Chemistry and Physics, 2017_

## Referee Comment (RC1) · Anonymous Referee #1 · 2 Jun 2017

**Review of Oscillations in atmospheric water above Switzerland**

*Klemens Hocke, Francisco Navas-Guzman, Lorena Moreira, Leonie Bernet, and Christian Mätzler*

**General comments**

Although the paper is based on a very valuable dataset (a continuous time series of CF, ILW, and IWV from the TROWARA at Bern for the period 2004-2016), and the data analysis methods used are certainly valid and well established, I am disappointed in the physical interpretation of the obtained results. The manuscript reads more as a summing up of findings, and often no explanations are given:

- On page 5, line 31: "it is surprising that the power spectrum of the zonal wind has strong annual harmonics reaching up to the fourth harmonic". Couldn't you try to explain this surprise?
- on page 6, lines 4-8: "The amplitude of the annual oscillation was strongest around 2010 to 2011. …. The semi-annual oscillation is strong from 2010 to 2014. … Similar to CF, the SAO in ILW is strong from 2010 to 2014." What is so specific about this 2010 to 2014 time period to explain the strongest amplitudes in the annual and semi-annual oscillations of those parameters?
- On page 6, lines 31-32: "The amplitude maxima are at a period of 7 days for CF, 6 days for ILW, 365 days for IWV, and 17 days for u". Can you relate the 6-7 and 17 day periods to atmospheric/weather event phenomena and what causes the difference between those periods for CF/ILW and u, given your earlier argument that those 3 variables are closely connected?
- On page 7, lines 2-3: "Figure 8 shows the mean amplitudes as function of the month and the period. The climatologies of CF, ILW and IWV show some similarities with increased amplitudes in the period range 5-10 days from spring to fall." Is this period related to synoptic scale weather events and why not in wintertime?

On the other hand, if the authors tried to explain some of the findings, their interpretation is too suggestive.

- on page 5, from line 32 onwards: "Since lower tropospheric wind is a major player for cloud formation and transport processes we suggest that the spectral components in the zonal wind spectrum are possibly the cause for the annual and semi-annual oscillations in the power spectra of CF and ILW". Did the authors investigate other causes?
- on page 6, lines 11-12: "It is surprising that the AO of the CF is almost in anti-phase to the AO in ILW which peaks in July. We think that convective cumulus clouds are responsible for the high ILW values in July". It is obvious that convective cumulus clouds form mostly in summer, but can

you make this thought more scientifically sounded (observations of clouds, CAPE index calculation etc.)?

- On the same page, the interpretation in lines 16-21 is also made on a lot of assumptions, which are not proven by the authors: "It is obvious that the climatology of u is rather similar to the climatology of CF in Figure 5. It seems that the strong eastward wind in December and January transports stratus cloud layers to Switzerland. Related to the study of Nuijens and Stevens (2012) we may argue that an increase in the lower tropospheric wind u leads to a deepening of the cloud layer. In addition, one may argue that an eastward advection of moist air from the Atlantic towards the Swiss plateau and the Alps occur which leads to a maximum of CF in winter". To my opinion, these statements might only be made after a careful cluster analysis of the trajectories of the air parcels arriving at Bern. I think that the entire paper would greatly benefit from such an analysis.
- On page 7, lines 3-4, it is stated: "The climatology of u shows a 20-day oscillation in winter which is possibly related to a Rossby wave". If you make such a statement, you should argue this.

Due to the lack of physical interpretation of the observed frequencies, the focus of the paper now lies on the demonstration of the analysis technique of identifying periods in the power spectra of CF, ILW, and IWV. The authors should investigate more time in the interpretation of the identified periods and the links between CF, ILW, IWV, u, and other variables, e.g. based on an identification of the origin of the air masses arriving at Bern. I would therefore suggest to make a major revision of the paper.

**Specific comments**

- On page 4, lines 29-30: "Finally, we like to mention that the CF, ILW and IWV measurements of TROWARA at Bern are representative for the Swiss plateau". Do you have references or arguments for this statement?
- Fig 2: the authors point to the similarity between the normalized power spectra of CF, ILW and u. This is certainly true for the two peaks around the fourth harmonic. But what about the rather strong frequency around 2.4 cycles/year in both the CF and ILW, which is not so prominent in the u?

**Technical corrections**

- Page 2, line 15: there are two "the" before seasonal
- Page 5, line 24: I would write "information" instead of "informations"
- Page 5, line 31: there are two "to" at the end of this line.

---

## Referee Comment (RC2) · Anonymous Referee #2 · 20 Jun 2017

The interesting data set of long-term microwave radiometer measurements used for the analysis gives a good opportunity for analyzing the variability of cloud parameters and IWV. However, the paper is very descriptive. There is only very little interpretation of the findings and when it is given it is often not proved well. Please give a more detailed interpretation of your findings and conclusions especially for the statements in the following:

- Why is it surprising that the power spectrum of the zonal wind has strong annual harmonics reaching up to the fourth harmonic? Give an explanation for this behavior. (p. 5 l. 31-32)
- What can you conclude from the similarity of SAO in ILW and SAO in CF? (p. 6 l. 7)
- Please explain your statement that convective cumulus clouds are responsible for the high ILW values in July. (p.6 l. 12)
- P. 6 l. 12-15: You explain which line shows what but you do not describe what you can see in the right panel of Fig. 5 and what you can conclude from that.
- To conclude a transport of air masses from the Atlantic from the zonal wind speed is speculative and should be proved.  (p.6 l. 20)
- How do you come to the conclusion that the 20 day-oscillation is related to a Rossby wave? Please explain.
- Your conclusions are mostly a summary. Here, an explanation of the connection between your findings about ILW, IWV and CF should be given.

There are also some minor corrections needed:

- Do not mention the "positive linear trend" of IWV in the abstract while the trend is not subject of this paper as you say on p.5 l.23.
- Please give a short explanation (1-2 sentences) how you determined the coefficients (p.4 l.1)
- Fig. 8: The description/caption of colors and lines is missing.
- Fig. 9: The description of the vertical lines is missing.

In my opinion it is inconvenient to give interpretations in the figure captions as in Fig. 6 "The seasonal change of u is similar …" and in Fig. 9:"While the weekly cycle in IWV…" Put these descriptions in the text instead of the figure captions.

The text could benefit from a rephrasing. For example, it would be nice to read an alternative to "Figure x shows…" and do not say "A spectral analysis of the green curves…" but "A spectral analysis of the monthly means…" (p. 5 l. 24).

Technical corrections:

- P.3 l. 10: remove brackets in citation
- P.3 l. 19: "where TB [is] the observed"
- P.5 l. 31: remove one of the two "to"

---

## Author Comment (AC1) · 13 Jul 2017

**Dear Editor, dear Reviewers,**

**We thank you for your constructive reviews. It was possible to consider almost all of your comments and to improve the article. Particularly we added a discussion of the synoptic weather types which lead to cloud formation over the Swiss plateau.**

**Point to point response:**

**Reviewer 1**

I am disappointed in the physical interpretation of the obtained results. The manuscript reads more as a summing up of findings, and often no explanations are given

We agree and we screened the literature for climatologies of cloud types. However we found only one paper (Scherrer and Appenzeller, 2014) which provides a climatology of fog and low stratus in the Swiss plateau. However, there are studies about the synoptic weather types in Switzerland which can be connected to cloud types (Collaud Coen et al., 2011; MeteoSwiss, 2015). So it was possible to add a physical discussion and interpretation in section 3 for a better understanding of the nature of the oscillations in atmospheric water parameters.
* * *
- On page 5, line 31: "it is surprising that the power spectrum of the zonal wind has strong annual harmonics reaching up to the fourth harmonic". Couldn't you try to explain this surprise?

The surprise was that the power do not exponentially decay with increase of the order of the harmonics. Further, the random nature of cyclones and anticylones at mid-latitudes would favour more white noise in the power spectra. We added a small discussion in the revised manuscript:

*text of the revised manuscript is always in cursive letters:*

*"It is surprising that the power spectrum of the zonal wind has strong annual harmonics reaching up to the fourth harmonic. Actually, one would assume only an annual oscillation in the prevailing westerly wind at northern midlatitudes which is larger during the winter than during the summer. The cyclones and anticyclones embedded in the westerly mean flow would be expected to have a random nature which would produce white noise in the spectrum. However, the $u$-spectrum in Fig.\ref{fig2} shows that there is an harmonic order in the temporal $u$-fluctuations favouring the occurrence of annual harmonics up to the fourth order. The harmonics may result from an interaction between the annual oscillation and intra-seasonal oscillations where the latter could be connected to synoptic-scale variations or synoptic weather types."*
* * *
- on page 6, lines 4-8: "The amplitude of the annual oscillation was strongest around 2010 to 2011. .... The semi-annual oscillation is strong from 2010 to 2014. ... Similar to CF, the SAO in ILW is strong from 2010 to 2014." What is so specific about this 2010 to 2014 time period to explain the strongest amplitudes in the annual and semi-annual oscillations of those parameters?

That's a good question. Actually we would need to analyze time series of cloud type frequencies in order to understand why the SAO was stronger in 2010-2014. We added a qualitative explanation how a forcing of the SAO by an increase of cumuliform clouds could happen:

*"A relationship between CF and ILW is expected since CF$=0$  if ILW $< 2.3$g/m$^2$.  An inter-annual change of the occurrence rate of certain weather types could explain the inter-annual variation of the SAO. For example, an enhancement in the occurrence of cumuliform clouds  in the summers from 2010 to 2014 may lead to the enhanced SAO  from 2010 to 2014. In future, the automated cloud type classification by thermal infrared cameras may provide objective time series of cloud type frequencies."*
* * *
- On page 6, lines 31-32: "The amplitude maxima are at a period of 7 days for CF, 6 days for ILW, 365 days for IWV, and 17 days for u". Can you relate the 6-7 and 17 day periods to atmospheric/weather event phenomena and what causes the difference between those periods for CF/ILW and u, given your earlier argument that those 3 variables are closely connected?

In the revised manuscript we discuss several causes for the generation of clouds. Though the variations in the zonal wind are related to advective weather types (east and west), they do not include the other weather types. Thus one cannot expect a full agreement of the wind and the cloud spectra.  In the revised manuscript we discuss the discrepancy of the u, CF, ILW and IWV spectra at short periods:

*"However, it is evident that the  climatology of the $u$ spectrum  cannot explain the  7 day-oscillation of CF, ILW and IWV during summer. This indicates that advective forcing is not the reason of the 7-day oscillation in summer. The 7 day-oscillation can be a  man-induced effect that may be enabled by periodic human activities during flat pressure gradient situations which prevail during summer \citep{coen2011}. The synoptic motion of the flat pressure gradient weather type is dominated by small-scale circulations."*
* * *
- On page 7, lines 2-3: "Figure 8 shows the mean amplitudes as function of the month and the period. The climatologies of CF, ILW and IWV show some similarities with increased amplitudes in the period range 5-10 days from spring to fall." Is this period related to synoptic scale weather events and why not in wintertime?

We selected the summer period since the 7-day oscillation is stronger during summer. The statistics show that the "flat pressure gradient" weather type (or also called convective indifferent) dominates during summer with convective forcing of cumuliform clouds.  So we suggest that the 7-day oscillation is a mode of the convective indifferent weather type.

*"The 7 day-oscillation can be a  man-induced effect that may be enabled by periodic human activities during flat pressure gradient situations which prevail during summer \citep{coen2011}. The synoptic motion of the flat pressure gradient weather type is dominated by small-scale circulations."*
* * *
1. On the other hand, if the authors tried to explain some of the findings, their interpretation is too suggestive.
   o  on page 5, from line 32 onwards: "Since lower tropospheric wind is a major player for cloud formation and transport processes we suggest that the spectral components in the zonal wind spectrum are possibly the cause for the annual and semi-annual oscillations in the power spectra of CF and ILW". Did the authors investigate other causes?

We agree our past interpretation was too short and simple. In the new version, we include possible other causes which are connected to the occurrence of different synoptic weather types and their seasonal change.

*"Since lower tropospheric wind is a major player for cloud formation and transport processes we suggest that the spectral components in the zonal wind spectrum could be one cause for the annual and semi-annual oscillations in the power spectra of CF and ILW. In addition the periodicities of 97 and 85 days (close to the*

*fourth harmonic) are strong in the spectra of u, CF and ILW. However, cloud formation also depends on synoptic weather types which often have a seasonal dependence. For example, the situation of a flat-pressure gradient weather type (or convective indifferent type) in West and Central Europe is typical for summer where convective forcing is often larger than advective forcing above Switzerland \citep{schlemmer2011,coen2011,meteoswiss2015}. The high evaporation rate during summer also supports that a moist atmosphere is getting unstable, and a diurnal convection cycle leads to cumuliform clouds in the afternoon and evening hours \citep{schlemmer2011,meteoswiss2015}.*

*During winter, the Swiss plateau often has low stratus which develops from condensation of atmospheric water vapour near to the cold Earth surface. Turbulence spreads the fog or cloud droplets up to the inversion layer in about 1.5 km altitude. \cite{scherrer2014} reported that 6-8 days per month in the Swiss plateau during winter have fog and stratus over an half day or more (e.g., low stratus before noon). Stratus in the Swiss plateau during winter is often associated with a cold wind from the north east which is called the Bise \citep{meteoswiss2015}. \cite{coen2011} reported that there is also a seasonal cycle of the advective weather types with an occurrence rate of about 45-50\% during winter and about 20\% in summer. Particularly, the warm and cold fronts of cyclones are passing Switzerland where the rising air masses at the warm front induce middle and high-level clouds. Further the north and the south foehn can be associated with cloud formation over the Swiss plateau. The occurrence of foehn is decreased during summer \citep{coen2011}.*
*Thus, the enhancement of cloud fraction by low stratus and advective weather types in winter and cumuliform clouds in summer may induce a semi-annual oscillation in CF and ILW over Bern."*
* * *
- on page 6, lines 11-12: "It is surprising that the AO of the CF is almost in anti-phase to the AO in ILW which peaks in July. We think that convective cumulus clouds are responsible for the high ILW values in July". It is obvious that convective cumulus clouds form mostly in summer, but can

you make this thought more scientifically sounded (observations of clouds, CAPE index calculation etc.)?

Yes, we explain in detail why there are more cumuliform clouds above the Swiss plateau during summer. This is based on weather type classifications presented by Collaud Coen et al. (2011).

*"However, cloud formation also depends on synoptic weather types which often have a seasonal dependence. For example, the situation of a flat-pressure gradient weather type (or convective indifferent type) in West and Central Europe is typical for summer where convective forcing is often larger than advective forcing above Switzerland \citep{schlemmer2011,coen2011,meteoswiss2015}. The high evaporation rate during summer also supports that a moist atmosphere is getting unstable, and a diurnal convection cycle leads to cumuliform clouds in the afternoon and evening hours \citep{schlemmer2011,meteoswiss2015}."*
* * *
- On the same page, the interpretation in lines 16-21 is also made on a lot of assumptions, which
  are not proven by the authors: "It is obvious that the climatology of u is rather similar to the climatology of CF in Figure 5. It seems that the strong eastward wind in December and January transports stratus cloud layers to Switzerland. Related to the study of Nuijens and Stevens (2012) we may argue that an increase in the lower tropospheric wind u leads to a deepening of the cloud layer. In addition, one may argue that an eastward advection of moist air from the Atlantic towards the Swiss plateau and the Alps occur which leads to a maximum of CF in winter". To my opinion, these statements might only be made after a careful cluster analysis of the trajectories of the air parcels arriving at Bern. I think that the entire paper would greatly benefit from such an analysis.

Yes, we slightly corrected and expanded our interpretation since the advective west weather type is mostly connected to the occurrence of middle and high-level clouds at the warm front of a cyclone. On the other hand we learned that low stratus is connected to the north easterly wind and to the cold Earth surface in winter.

*"During winter, the Swiss plateau often has low stratus which develops from condensation of atmospheric water vapour near to the cold Earth surface. Turbulence spreads the fog or cloud droplets up to the inversion layer in*

*about 1.5 km altitude. \cite{scherrer2014} reported that 6-8 days per month in the Swiss plateau during winter have fog and stratus over an half day or more (e.g., low stratus before noon). Stratus in the Swiss plateau during winter is often associated with a cold wind from the north east which is called the Bise \citep{meteoswiss2015}. \cite{coen2011} reported that there is also a seasonal cycle of the advective weather types with an occurrence rate of about 45-50\% during winter and about 20\% in summer. Particularly, the warm and cold fronts of cyclones are passing Switzerland where the rising air masses at the warm front induce middle and high-level clouds. Further the north and the south foehn can be associated with cloud formation over the Swiss plateau. The occurrence of foehn is decreased during summer \citep{coen2011}."*
* * *
- On page 7, lines 3-4, it is stated: "The climatology of u shows a 20-day oscillation in winter which is possibly related to a Rossby wave". If you make such a statement, you should argue this.

We agree. We added a reference and some more details for our suggestion:

*"The climatology of the $u$ spectrum shows a 20 day-oscillation in winter which is possibly related to a Rossby wave. The 20-day period is close to 16 days which is a theoretical period of a normal mode of a free Rossby wave with a westward-propagating zonal wavenumber 1 \citep{sassi2012}."*
* * *
1. The authors should investigate more time in the interpretation of the identified periods and the links between CF, ILW, IWV, u, and other variables, e.g. based on an identification of the origin of the air masses arriving at Bern. I would therefore suggest to make a major revision of the paper.

We think that the discussion of the synoptic weather types is a good way to understand the origin of the air and how the cloud formation in the Swiss plateau works. Transport studies of moist air is beyond our capabilities.
* * *
1. **Specific comments**
   - On page 4, lines 29-30: "Finally, we like to mention that the CF, ILW and IWV measurements of TROWARA at Bern are representative for the Swiss plateau". Do you have references or arguments for this statement?

We tell now:

*"Finally, we like to mention that the CF, ILW and IWV measurements of TROWARA at Bern are within the central basin of the Swiss plateau."*
* * *
- Fig 2: the authors point to the similarity between the normalized power spectra of CF, ILW and u. This is certainly true for the two peaks around the fourth harmonic. But what about the rather strong frequency around 2.4 cycles/year in both the CF and ILW, which is not so prominent in the u?

We mention now that there is also a discrepancy. We don't know the reason possibly the u-variations are only one cause for cloud occurrence:

*"However, the component at 150 days only occur in CF and ILW. "*
* * *
1. **Technical corrections**

- o   Page 2, line 15: there are two "the" before seasonal
- o   Page 5, line 24: I would write "information" instead of "informations"
- o   Page 5, line 31: there are two "to" at the end of this line.

Thank you, we added your corrections in the new manuscript!
Thank you for your thoroughful review which significantly improved our study!

**Point to point response:**

First at all, there are many similar remarks of Reviewer 1 and Reviewer 2 which show that it is wise to consider them.

**Reviewer 2**

*However, the paper is very descriptive. There is only very little interpretation of the findings and when it is given it is often not proved well. Please give a more detailed interpretation of your findings and conclusions especially for the statements in the following:*

We agree and we added a discussion on synoptic weather types, their seasonal change and their relation to cloud types in section 3 Results.
* * *
- • *Why is it surprising that the power spectrum of the zonal wind has strong annual harmonics reaching up to the fourth harmonic? Give an explanation for this behavior. (p. 5 l. 31-32)*

The surprise was that the power do not exponentially decay with increase of the order of the harmonics. Further, the random nature of cyclones and anticylones at mid-latitudes would favour more white noise in the power spectra.  We added a small discussion in the revised manuscript:

*text of the revised manuscript is always in cursive letters:*

"*It is surprising that the power spectrum of the zonal wind has strong annual harmonics reaching up  to the fourth harmonic. Actually, one would assume only an annual oscillation in the prevailing westerly wind at northern midlatitudes which is larger during the winter  than during the summer. The cyclones and anticyclones embedded in the westerly mean flow would be expected to have a random nature which would produce white noise in the spectrum. However, the $u$-spectrum in  Fig.\ref{fig2} shows that there is an harmonic order in the temporal $u$-fluctuations favouring the occurrence of annual harmonics up to the fourth order. The harmonics may result from an interaction between the annual oscillation and intra-seasonal oscillations  where the latter could be  connected to synoptic-scale variations or synoptic weather types.*"
* * *
- • *What can you conclude from the similarity of SAO in ILW and SAO in CF? (p. 6 l. 7)*

We suggest now that the seasonal change of synoptic weather types produce the SAO with cumuliform clouds in summer and stratus and middle/high clouds in winter.

*However, cloud formation also depends on synoptic weather types which often have a seasonal dependence. For example, the situation of a flat-pressure gradient weather type  (or convective indifferent type) in West and Central Europe  is typical for summer  where convective forcing is often larger than advective forcing above Switzerland  \citep{schlemmer2011,coen2011,meteoswiss2015}. The high evaporation rate during summer also supports that a moist atmosphere is getting unstable, and a diurnal convection cycle leads to cumuliform clouds in the afternoon and evening hours \citep{schlemmer2011,meteoswiss2015}.*

*During winter, the Swiss plateau often has low stratus which develops from condensation of atmospheric water vapour near to the cold Earth surface. Turbulence spreads the fog or cloud droplets up to the inversion layer in about 1.5 km altitude. \cite{scherrer2014} reported that 6-8 days per month in the Swiss plateau during winter have fog and stratus over an half day or more (e.g., low stratus before noon). Stratus in the Swiss plateau during winter is often associated with a cold wind from the north east which is called the Bise \citep{meteoswiss2015}. \cite{coen2011} reported that there is also a seasonal cycle of the advective weather types with an occurrence rate of about 45-50\% during winter and about 20\% in summer. Particularly, the warm and cold fronts of cyclones are passing Switzerland where the rising air masses at the warm front induce middle and high-level clouds. Further the north and the south foehn can be associated with cloud formation over the Swiss plateau. The occurrence of foehn is decreased during summer \citep{coen2011}.*
   *Thus, the enhancement of cloud fraction by low stratus and advective weather types in winter and cumuliform clouds in summer may induce a semi-annual oscillation in CF and ILW over Bern.*
* * *
- Please explain your statement that convective cumulus clouds are responsible for the high ILW values in July. (p.6 l. 12)

*"For example, the situation of a flat-pressure gradient weather type (or convective indifferent type) in West and Central Europe is typical for summer where convective forcing is often larger than advective forcing above Switzerland \citep{schlemmer2011,coen2011,meteoswiss2015}. The high evaporation rate during summer also supports that a moist atmosphere is getting unstable, and a diurnal convection cycle leads to cumuliform clouds in the afternoon and evening hours \citep{schlemmer2011,meteoswiss2015}."*
* * *
- P. 6 l. 12-15: You explain which line shows what but you do not describe what you can see in the right panel of Fig. 5 and what you can conclude from that.

*We agree and added a small discussion:*

*"The right-hand-side panels show the mean behaviour of the combined AO and SAO in black while the green lines show the mean behaviour derived from the monthly mean series of CF, ILW and IWV. In addition the standard error of the mean is given by green error bars.*
 *We can see that the AO and the SAO component fit a major part of the observed monthly mean series. There are only a few month-to-month variations in the climatology of monthly means (green curve) which are not approximated by the combined AO and SAO (black curve)."*
* * *
- To conclude a transport of air masses from the Atlantic from the zonal wind speed is speculative and should be proved. (p.6 l. 20)

Yes, we added a discussion. Generally it is known that the cyclones and anticyclones are mostly coming from the Atlantic drifting by the prevailing eastward wind to Europe.

*"It seems that the strong eastward wind in December and January is associated with the advective weather type which generates middle and high clouds over Switzerland in the warm zone and the warm front of cyclones \citep{meteoswiss2015}. Related to the study of \cite{nuijens2012} we argue that an increase in the lower tropospheric wind $u$ leads to a deepening of the cloud layer. In addition, one may argue that an eastward advection of moist air from the Atlantic towards the Swiss plateau and the Alps occur which leads to a maximum of CF in winter. The so-called advective west (AW) weather type is enhanced by about 10\% during winter compared to summer \citep{coen2011}. Generally the sum of the advective weather types have an occurrence rate of about 45-50\% during winter while they are below 25\% during summer \citep{coen2011}."*
* * *
- How do you come to the conclusion that the 20 day-oscillation is related to a Rossby wave? Please explain
   We agree. We added a reference and some more details for our suggestion:

*"The climatology of the $u$ spectrum shows a 20 day-oscillation in winter which is possibly related to a Rossby wave. The 20-day period is close to 16 days which is a theoretical period of a normal mode of a free Rossby wave with a westward-propagating zonal wavenumber 1 \citep{sassi2012}."*
* * *
- Your conclusions are mostly a summary. Here, an explanation of the connection between your findings about ILW, IWV and CF should be given.

We agree and we renamed the section as Summary. Nevertheless, we added some conclusions:

*"The semi-annual oscillations (SAO) of CF and ILW are strong from 2010 to 2014. We suggest that the SAO could be related to the occurrence frequency of certain weather types which lead for example to low stratus in winter and cumuliform clouds in summer. In future, we expect that automated cloud classification by thermal infrared cameras may give us climatologies of cloud types which could be helpful for interpretation of the periodicities in CF and ILW."*
* * *
- Do not mention the "positive linear trend" of IWV in the abstract while the trend is not subject of this paper as you say on p.5 l.23.

We agree and we removed the trend sentence.
* * *
- Please give a short explanation (1-2 sentences) how you determined the coe=cients (p.4 l.1)

It is not so easy to explain the method in short. However we changed the formulations and mention that forward modelling of the opacities is necessary. For more details the reader has to consult the references.

*" ... where the coefficients $a''$ and $b''$ are not really constant since they can partly depend on air pressure. \cite{maetzler2009} show that these coefficients can be statistically derived by means of nearby radiosonde measurements and fine-tuned at times of periods with a clear atmosphere. The radiosonde yields the atmospheric profile which is used for forward modelling of the brightness temperatures and opacities which would have been observed by TROWARA. Further, the radiosonde provides IWV so that the equation set ( \ref{eq:tau2}) can be solved for the coefficients $a''$ and $b''$ for clear sky \citep{maetzler2009}. The coefficient $c''$ is the mass absorption coefficient of cloud water. It depends on temperature (and frequency), but not on pressure. It is derived from the physical expression of Rayleigh absorption by clouds \citep{maetzler2009}. The equation set ( \ref{eq:tau2}) permits the retrieval of IWV and ILW if the opacities are measured at 21 and 31 GHz. Thus, a dual channel microwave radiometer can monitor IWV and ILW with a time resolution of 6-11 seconds and nearly all-weather capability during day and nighttime."*
* * *
- Fig. 8: The description/caption of colors and lines is missing.

We adjusted Figure 8 so that the color bars are well described and the contour lines are removed.
* * *
- Fig. 9: The description of the vertical lines is missing.

*"Weekday 1 corresponds to Sunday, weekday 2 corresponds to Monday, and so on. The vertical lines indicate the error of the mean of the averaged values."*
* * *
1. In my opinion it is inconvenient to give interpretations in the &gure captions as in Fig. 6 "The seasonal change of u is similar ..." and in Fig. 9:"While the weekly cycle in IWV..." Put these descriptions in the text instead of the &gure captions.

We agree and we removed all the interpretations from the figure captions.
* * *
1. The text could bene&t from a rephrasing. For example, it would be nice to read an alternative to "Figure x shows…" and do not say "A spectral analysis of the green curves…" but "A spectral analysis of the monthly means…" (p. 5 l. 24).

We agree and we sometimes use "depicts" instead of "shows". We also  the "monthly mean series" instead of "green curves".
* * *
1. Technical corrections:
   - P.3 l. 10: remove brackets in citation
   - P.3 l. 19: "where TB [is] the observed"
   - P.5 l. 31: remove one of the two "to"

We agree and we inserted your technical corrections in the new manuscript.
   *Thank you for your help!*

*References:*

Collaud Coen, M., Weingartner, E., Furger, M., Nyeki, S., Prévôt, A. S. H., Steinbacher, M., and Baltensperger, U.: Aerosol climatology and planetary boundary influence at the Jungfraujoch analyzed by synoptic weather types, Atmospheric Chemistry and Physics, 11, 5931– 5944, doi:10.5194/acp-11-5931-2011, 2011.

Mätzler, C. and Morland, J.: Refined Physical Retrieval of Integrated Water Vapor and Cloud Liquid for Microwave Radiometer Data, IEEE Transactions on Geoscience and Remote Sensing, 47, 1585–1594, 2009.

MeteoSwiss: Typische Wetterlagen im Alpenraum, Report of Bundesamt für Meteorologie und Klimatologie MeteoSchweiz, pp. 1–28, http://www.meteoswiss.admin.ch/content/dam/meteoswiss/de/service-und-publikationen/Publikationen/doc/Web_Wetterlagen_ DE_low.pdf, 2015.

Nuijens, L. and Stevens, B.: The Influence of Wind Speed on Shallow Marine Cumulus Convection, Journal of the Atmospheric Sciences, 69, 168–184, doi:10.1175/JAS-D-11-02.1, 2012.

Sassi, F., Garcia, R. R., and Hoppel, K. W.: Large-Scale Rossby Normal Modes during Some Recent Northern Hemisphere Winters, Journal of the Atmospheric Sciences, 69, 820–839, doi:10.1175/JAS-D-11-0103.1, 2012.

Scherrer, S. C. and Appenzeller, C.: Fog and low stratus over the Swiss Plateau - a climatological study, International Journal of Climatology, 34, 678–686, doi:10.1002/joc.3714, 2014.

Schlemmer, L., Hohenegger, C., Schmidli, J., Bretherton, C. S., and Schär, C.: An Idealized Cloud-Resolving Framework for the Study of Midlatitude Diurnal Convection over Land, Journal of the Atmospheric Sciences, 68, 1041–1057, doi:10.1175/2010JAS3640.1, 2011.